# Healthcare-Associated Laboratory-Confirmed Bloodstream Infections—Species Diversity and Resistance Mechanisms, a Four-Year Retrospective Laboratory-Based Study in the South of Poland

**DOI:** 10.3390/ijerph18052785

**Published:** 2021-03-09

**Authors:** Agnieszka Chmielarczyk, Monika Pomorska-Wesołowska, Dorota Romaniszyn, Jadwiga Wójkowska-Mach

**Affiliations:** 1Department of Microbiology, Faculty of Medicine, Jagiellonian University Medical College, 31-121 Krakow, Poland; agnieszka.chmielarczyk@uj.edu.pl (A.C.); dorota.romaniszyn@uj.edu.pl (D.R.); 2Department of Microbiology, Analytical and Microbiological Laboratory of Ruda Slaska, KORLAB NZOZ, 41-700 Ruda Slaska, Poland; monikapw@op.pl

**Keywords:** antibiotic resistance, laboratory-confirmed bloodstream-infections, species diversity

## Abstract

Introduction: Regardless of the country, advancements in medical care and infection prevention and control of bloodstream infections (BSIs) are an enormous burden of modern medicine. Objectives: The aim of our study was to describe the epidemiology and drug-resistance of laboratory-confirmed BSI (LC-BSIs) among adult patients of 16 hospitals in the south of Poland. Patients and methods: Data on 4218 LC-BSIs were collected between 2016–2019. The identification of the strains was performed using MALDI-TOF. Resistance mechanisms were investigated according to European Committee on Antimicrobial Susceptibility Testing, EUCAST recommendations. Results: Blood cultures were collected from 8899 patients, and LC-BSIs were confirmed in 47.4%. The prevalence of Gram-positive bacteria was 70.9%, Gram-negative 27.8% and yeast 1.4%. The most frequently isolated genus was *Staphylococcus* (50% of all LC-BSIs), with a domination of coagulase-negative staphylococci, while *Escherichia coli* (13.7%) was the most frequent Gram-negative bacterium. Over 4 years, 108 (2.6%) bacteria were isolated only once, including species from the human microbiota as well as environmental and zoonotic microorganisms. The highest methicillin resistant *Staphylococcus aureus* (MRSA) prevalence was in intensive care units (ICUs) (55.6%) but *S. aureus* with resistance to macrolides, lincosamides and streptogramins B (MLS_B_) in surgery was 66.7%. The highest prevalence of *E. faecalis* with a high-level aminoglycoside resistance (HLAR) mechanism was in ICUs, (84.6%), while *E. faecium*-HLAR in surgery was 83.3%. All cocci were fully glycopeptide-sensitive. Carbapenem-resistant Gram-negative bacilli were detected only in non-fermentative bacilli group, with prevalence 70% and more. Conclusions: The BSI microbiology in Polish hospitals was similar to those reported in other studies, but the prevalence of MRSA and enterococci-HLAR was higher than expected, as was the prevalence of carbapenem-resistant non-fermentative bacilli. Modern diagnostic techniques, such as MALDI-TOF, guarantee reliable diagnosis.

## 1. Introduction

Community-onset (CO) and healthcare-associated bloodstream infections (BSIs) are major causes of severe febrile illness and death worldwide. In hospitals, BSIs are the most common healthcare-associated infections (HAI) with significant mortality and additional costs. In European Union (EU) countries, the attributable mortality of intensive care unit (ICU)-acquired BSIs was estimated at 5.0%, and the attributable excess length of ICU stay was at 14 days, accounting for 4505 deaths as the direct consequence of the infection and 1.26 million days of excess ICU stay in EU acute care hospitals every year [1]. Regardless of the region of the world, the advancement of medical care and the advancement of surveillance of infections, both primary and secondary BSIs are an enormous burden of modern medicine [2].

The accurate identification of the source of infection and determination of the etiological agent are important for the optimization of therapy, including targeted antibiotic therapy, which can contribute to improved prognosis in patients with BSIs. Culturing blood samples in automated systems such as BACTEC (Becton Dickinson) is still the basic test in classic microbiological diagnostics. Yet, many blood cultures are negative and the empirical treatment of culture-negative BSIs is of great interest [3]. Molecular tests based on polymerase chain reaction (PCR) and fluorescent in situ hybridization (FISH) can help increase the sensitivity of detecting bacteria in the blood and also speed up this stage, but they are still not routine diagnostics in Poland [4].

Recently, however, in laboratory confirmed bloodstream infections (LC-BSIs), it has been possible to identify microorganisms grown from blood faster and faster, thanks to the new and increasingly common MALDI-TOF (matrix-assisted laser desorption/ionization time-of-flight) identification system. In addition to quick identification, this system also provides accurate species diagnostics, which is important in the selection of empirical therapy. As shown by the research of the European Society of Clinical Microbiology and Infectious Disease (ESCMID) Study Group for BSI, Endocarditis and Sepsis (ESGBIES), 2/3 of laboratories apply rapid technologies to process positive blood cultures but the remaining 1/3 are still basing their techniques on classical methods and only 13% had established a 24 h service [5].

In recent years, the epidemiology of BSIs in Europe has indicated a change in the prevalence of its microbial etiological agents. The European Centre for Disease Prevention and Control (ECDC) report for 2008–2012 indicates the dominance of Gram-positive bacteria (mainly coagulase-negative staphylococci; CNS) and Candida, and in 2015–2017, Gram-positive bacteria also dominated, but a significantly increasing share of *E.coli* (increase from 7% to 9%), *Klebsiella* spp. (increase from 7% to 12%), *Enterobacter* spp. (increase from 5% to 8%) and *Pseudomonas* spp. (increase from 7% to 9%) can be seen [1]. In Poland, in particular in primary BSIs, CNS also dominates (depending on the study, from 20.8% to 44.2%), however, in secondary BSIs, there is still a large share of bacteria of the genus *Acinetobacter* (from 17.3% to 34.5%), although at present this trend has stopped somewhat. The abundant presence of Gram-negative species in BSIs, such as *Acinetobacter baumannii* or *Klebsiella pneumoniae*, is also associated with high antimicrobial resistance [6,7,8,9]. In BSIs, early active antibiotic therapy is associated with better outcomes. Therefore, international guidelines recommend empiric broad-spectrum antibiotics for all patients with suspected sepsis. However, both inadequate and unnecessarily broad empiric antibiotics were associated with higher mortality. Thus, it is essential to study the epidemiology of antibiotic-resistant pathogens in culture-positive BSI patients and to estimate the risk of inadequate and unnecessarily broad antibiotic treatments in hospitals [10,11].

There is a lack of a multicenter systematic study of bloodstream infections in Poland and there are no such studies from central Europe; in addition, their microbiology and drug sensitivity are not static but constantly changing over time, necessitating continuous surveillance. Due to these reasons, the aim of our study is to describe the epidemiology and microbiology of bloodstream infections (BSIs) among adult patients of hospitals in the south of Poland.

## 2. Materials and Methods

The collected data came from the large microbiological laboratory KORLAB Ruda Slaska serving the Silesian Voivodeship (the south of Poland). Data on LC-BSI from four consecutive years between 1 January 2016 and 31 December 2019 were collected from single-profile and multi-profile hospitals and, in addition, data from surgical units (23 in total) and intensive care units (ICUs) (3 in total) were selected. Non-duplicate positive blood cultures from 8899 individual patients were included in the analysis (duplicate identical test results from the same patient were rejected).

### 2.1. Culture and Identification

All blood specimens of at least 8 mL were injected into aerobic and anaerobic blood culture bottles (BACTEC Lytic/10/Anaerobic/F and BACTEC Peds Plus, Becton Dickinson, Poland). Samples were taken in duplicate (right and left hand) from approximately 70% of the patients. The remaining 30% of patients were sampled in one replicate because, for various reasons (including the patient’s condition), it was not possible to collect more samples. Then, material was cultured on MacConkey agar, Columbia agar (at 37 °C, each for 24 h) and chocolate agar (at 37 °C, 48 h) (all from Becton Dickinson, Warszawa, Poland). The identification of the bacterial strains was performed by the MALDI-TOF automated identification system (Biotyper Bruker Daltonics, Leiderdorp, The Netherlands).

### 2.2. Resistance Mechanisms

Mechanisms of resistance were investigated according to EUCAST recommendations. A variety of methods were used. In some cases, tests were performed with two different methods.

Methicillin resistant *Staphylococcus aureus* (MRSA)/MRSE/methicillin resistant coagulase-negative staphylococci (MRCNS)-methicillin/oxacillin resistance was detected both phenotypically by MIC (minimum inhibitory concentration) determination of cefoxitin and oxacillin and by MALDI-TOF—through the identification of the PSM-*mec* peak. PSM-*mec* is a surface peptide and the proportion of MRSA with PSM-*mec* is variable, depending on regional epidemiology. The absence of this peak does not mean that the respective strain is not MRSA, but, if detected, it can be reliably considered as MRSA [12].

MLS_B_ (resistance to macrolides, lincosamides and streptogramins) along with the remaining drug resistance were determined in the MIDITECH apparatus (BEL-MIDITECH s.r.o, Bratislava, Slovak Republic) and phenotypically by disk diffusion tests with clindamycin (2 µg) and erythromycin (15 µg).

ESBL and AmpC—the production of extended-spectrum β-lactamase and the production of acquired AmpC β-lactamase—was detected in the MIDITECH apparatus (BEL-MIDITECH s.r.o, Slovak Republic) and phenotypically by disk diffusion tests with amoxicillin with clavulanic acid (20/10 µg), cefotaxime (30 µg), ceftazidime (30 µg), cefepime (30 µg), and cefoxitin (30 µg). ESBL detection and characterization is recommended or mandatory for infection control purposes.

HLAR (detection of high-level resistance to aminoglycosides) was detected phenotypically by disk diffusion tests with gentamicin (30 µg).

VRE (vancomycin resistance) were detected by MIC determination with an E-test.

Carbapenem resistance for imipenem, meropenem and ertapenem was tested together with the remaining drug resistance in the MIDITECH apparatus by determining the MIC. In the case of carbapenem-resistant strains, the mechanism of KPC (*Klebsiella pneumoniae* carbapenemase) was investigated using disk diffusion tests with a disk containing meropenem and a disk containing an inhibitor (boronic acid); the mechanism of MBL (metallo-β-lactamases) was investigated using disk diffusion tests with a disk containing imipenem, a disk with ceftazidime and a disk with an inhibitor (EDTA, ethylenediaminetetraacetic acid). Temocillin high-level resistance (MIC > 128 mg/L) was used as a phenotypic marker for OXA-48-like carbapenemase producers. The Carba NP (Carbapenemase Nordmann-Poirel) test for detection of carbapenem hydrolysis was performed as a confirmation test.

In this publication, multidrug resistance (MDR) stands for resistance to two or more groups of antibiotic classes (beta lactams, aminoglycosides, fluoroquinolones, etc.).

This work was approved by the Bioethics Committee of Jagiellonian University (1072.6120.50.2017). All data analyzed during this study were anonymized prior to analysis. The study was based on the data gathered during routine caretaking and the analysis did not include any individual participants’ data. As a result, no statements of consent from participants were required. The study in its present form was approved by the local Bioethics Committee of Jagiellonian University.

## 3. Results

From 1 January 2016 to 31 December 2019, blood cultures were performed on 8899 patients including 557 (6.3%) patients from surgical units and 363 (4.1%) patients from ICUs. In total, there were 4218 LC-BSIs (47.4%), 382 in surgical units (68.6%) and 279 in ICUs (76.9%) (Table 1). The percentage of LC-BSI bottle blood culture positivity rate increased over the following years, from 42% to 50.8% (respectively in 2016 and 2019).

Considering the total LC-BSIs (*n* = 4218, 100%), the presence of Gram-positive bacteria was demonstrated in 70.9% of cases, Gram-negative bacteria in 27.8% and yeast/fungi in 1.4%. In ICUs, a higher prevalence of Gram-negative bacteria (31.5%) and yeast/fungi (5.4%) was demonstrated compared to other departments, while in surgical units, only a higher proportion of yeast/fungi was found (3.9%) (Table 2).

In total, over 4 years, 86 different genera were identified, including 227 species of microorganisms (Table 3 and Appendix A). The greatest numbers of different Gram-positive bacterial species were found in the genus *Streptococcus* (21 species), *Corynebacterium* (20 species) and *Staphylococcus* (19 species), while among the Gram-negative bacteria, it was the genus *Acinetobacter* (7 species).

The species differentiation in *Corynebacterium* is interesting here, because these bacteria were detected within four years only in 98 cases of LC-BSI (i.e., much less frequently than in the case of the genus *Staphylococcus*: 2121 cases of LC-BSI and the genus *Streptococcus*: 261 cases of LC-BSI).

The species differentiation is greater in other units than in ICUs and surgical units; however, it is related to the much higher number of LC-BSI in these departments.

Thirty-six species of microorganisms were isolated more than 10 times over the study period. The most often isolated genus of Gram-positive bacteria was *Staphylococcus*. Over 4 years, these bacteria were confirmed in BSI in 2121 cases (50%). The largest share was found in coagulase-negative staphylococci: *Staphylococcus epidermidis* (19%) and *Staphylococcus hominis* (14%). Among Gram-negative bacteria, *Escherichia coli* was detected most frequently, 13.7%. The genus *Streptococcus*, *S. aureus* and the genus *Klebsiella* were isolated less frequently (6%, 6.5% and 6%, respectively (Table 4)).

One hundred and eight species of microorganisms were isolated only once over 4 years, including 70 species of Gram-positive bacteria, 34 species of Gram-negative bacteria and 4 species of yeast/fungi. Among them, we can distinguish species belonging to the human microflora and opportunistic pathogens causing infections mainly in patients with immunodeficiency, as well as environmental species and animals. Selected rare pathogens (rarely found in BSIs) isolated one or more times during the period studied are summarized in Table 5.

The mechanisms of resistance in Gram-positive and Gram-negative bacteria have been investigated. *Enterococcus faecalis* with the HLAR mechanism was more frequently found in ICUs (84.6%), while *Enterococcus faecium* with HLAR as well as combined HLAR and VRE was more often found in surgical units, 83.3% and 50%, respectively. Methicillin-resistant *S. aureus* strains as well as *S. aureus* strains with the MLS_B_ mechanism and with both resistance mechanisms were more often isolated in both ICUs (55.6%, 55.6% and 44.4%) and surgical units (33.3%, 66.7% and 33.3%). Both *S. aureus* and all CNS were fully sensitive to glycopeptides.

In the case of Gram-negative bacteria in ICUs, we deal significantly more often with carbapenem-resistant non-fermenting bacilli: *A. baumannii* (70.6%) and *Pseudomonas aeruginosa* (66.7%). Carbapenem-resistant *A. baumannii* (100%) and ESBL-producing *E. cloacae* (100%) were more frequently found in surgical departments (Table 6).

## 4. Discussion

BSI is one of the most important challenges of modern medicine. In EU countries, it is the second the most common HAI in ICUs. In 2008–2012, it was reported in 3.5% of ICU patients with a hospital stay of more than two days. The incidence rate was 5.2 episodes per 1000 patient-days for both primary and secondary BSIs [1]. Unfortunately, in Poland, the incidence of BSI (primary and secondary) in 2013–2015 was much higher: 9.2 per 1000 patient-days [6].

The etiological agents of healthcare-associated-BSI in Poland and the EU countries were also different. For example, according to ECDC data, the most frequently isolated microorganisms in ICU-acquired bloodstream infections were coagulase-negative staphylococci (CNS) (25.6%), whereas in Poland, *Staphylococcus aureus* displayed the prevalence of 13.4%, while the prevalence of CNS was 10.4% [1,6]. Fortunately, the present results are close to the European data, with the expected domination of CNS. On the other hand, in EU countries, the overall prevalence of *Candida* species was 8.2%, with the highest share of *C. albicans*, but in our data, far fewer yeasts were found in BSIs in ICUs.

According to ECDC data from 2017, the share of CNS for ICUs is 23.6%, while in our study, their share in ICUs is greater—35% [13], which is similar to the data from the USA from 20 years ago (1995–2002), where the share of CNS in ICUs also exceeded 1/3 of all isolated microorganisms. However, in non-ICUs in the US, CNS were isolated almost twice less frequently than in the present data [14]. On the basis of the data we have, we are not able to conclude whether it is related to pre-laboratory errors in blood sampling or whether it indicates a significant relationship between BSIs and the use of catheterization procedures [15,16].

The current gold standard for diagnostic of BSIs is the blood culture, but according to the ESCMID Study Group for Bloodstream Infections, Endocarditis and Sepsis (ESGBIES) study (2016–2017) from 28 countries and 209 laboratories, only 1/3 of laboratories used the classical processing of positive blood cultures, while two-thirds applied rapid technologies. MALDI-TOF from briefly incubated sub-cultures on solid media was the most commonly used approach to rapid pathogen identification from positive blood cultures—in 37.3% of laboratories, and direct disk diffusion was the most common rapid antimicrobial susceptibility testing method from positive blood culture. In the same study, the reported mean bottle blood culture positivity rate was only 13.8%. Thus, as the authors conclude, the laboratories have started to implement novel technologies for rapid identification for positive blood cultures, but current practices of blood culture diagnostics do not guarantee an optimal BSI management [5]. On the other hand, according to the National Healthcare Safety Network (NHSN) data, in an overall number of 85,994 (23.5%) central line-associated bloodstream infections, each was associated with an average of 1.1 reported pathogens [11].

In Polish laboratories, thanks to the method of identifying microorganisms using the MALDI-TOF system, the identification of microorganisms up to the species level has significantly improved, and thus the species diversity among LC-BSIs has increased. In our study, it was particularly noticeable in the case of *Corynebacterium* species. When *Corynebacterium* species were diagnosed using the API Coryne biochemical system, the identification of individual species was not always so accurate. Correct and precise species identification in the case of *Corynebacterium jeikeium* is important for the therapeutic process as it is a species resistant to a variety of antimicrobial agents. Vancomycin is recommended in empirical treatment [17,18]. Moreover, the detection and identification of *Stenotrophomonas maltophilia* from blood cultures is important because it is a species sensitive only to sulfamethoxazole/trimethoprim (SXT). In a situation in which the patient cannot receive SXT (e.g., due to intolerance) or the strain is also resistant to SXT, the therapy is problematic. Alternative therapies are then considered including tigecycline, colistin, and ticarcillin plus clavulanate [19]. In the case of fungi, it is important to correctly identify *Candida krusei*, a species that is naturally resistant to fluconazole, and *Candida glabrata*, which has a reduced sensitivity to fluconazole—this drug should not be used in therapy. *Candida lusitaniae*, on the other hand, is characterized by a reduced sensitivity to amphotericin B [20,21].

In recent years, in the literature, there has been a progressive increase in the number of reports of clinical cases associated with the isolation of various rare pathogens. Over the four years of our study, a large number of bacterial species that are such pathogens have been isolated and identified as a cause of BSI. Many of them are bacteria of the human skin microflora such as *Dermabacter hominis* or *Cutibacterium avidum* or intestinal microflora, where Gram-positive anaerobic cocci (GPAC) are an important group: *Peptoniphilus harei, Finegoldia magna, Anaerococcus hydrogenalis, Anaerococcus lactolyticus, AtopobiumrimaeRuminococcusgnavus* and other species such as *Solobacterium moorei*. These are opportunistic pathogens isolated primarily from patients with immunodeficiency or coexisting diseases. GPAC-bacteraemia is reported with increasing frequency and it is a condition with significant mortality, especially in the elderly [22,23]. Among the opportunistic pathogens derived from the human microflora, we can also distinguish bacteria of the oral cavity: *Rothia dentocariosa, Eikenella corrodens, Actinomyces odontolyticus* or *Actinomyces radicidentis*. Oral infections or immunosuppression may be the primary source or cause of bacteremia caused by these organisms [24,25].

It is worth noting that some non-pathogenic species of human microflora are isolated from BSIs in immunocompetent patients: *Gemella morbillorum* was isolated from a healthy 5-month-old baby [26], *Pantoea dispersa* from an immunocompetent 38-year-old woman [27], *Rothia mucilaginosa* from a 9-month-old baby [28], and *Brevibacterium paucivorans* from an older patient [29]. Some species described so far as rare pathogens emerge more and more frequently in BSI and are becoming significant human pathogens, such as *Kocuria rhizophila* isolated from catheter-related BSIs [30,31]. Others, such as environmental *Aeromonas hydrophila*, although rare, can cause severe or problematic infections due to drug resistance [32,33].

In our four-year study, there were also types of environmental bacteria that are described as exceedingly rare in human infections, such as *Paenibacillus, Aureimonas, Zimmermanella* [34,35,36]. Unfortunately, however, we do not have information on the clinical status of patients from whom these species have been isolated. Among the environmental strains identified by us, the genus *Microbacterium* is worth noting; Alonso et al. described a hospital outbreak caused by these bacteria including five cases of symptomatic bacteremia [37].

From the bacteria that are pathogens or the physiological flora of animals that we noted in our study, there are, among others, *Capnocytophaga canimorsus* which is canine and feline oral flora and *Erysipelothrix rhusiopathiae* isolated from numerous animals. These species most commonly cause BSI due to bites [38,39].

Comparing the BSIs in patients from ICUs and other hospital departments, our study shows a higher proportion of *Enterococcus* and Gram-negative bacteria from the genera *Acinetobacter* and *Klebsiella*, as well as *Candida* fungi. Bacteria isolated from ICUs are characterized by higher antimicrobial resistance. In our study, we did not analyze the full antimicrobial resistance profile but the main mechanisms of resistance in those species that have the highest share in BSI or are a significant therapeutic problem due to their drug resistance.

The highest resistance was found among Gram-positive microorganisms from ICUs in *E.faecalis* with the HLAR mechanism—84.6% vs. 42.1% of *E. faecalis* resistant to aminoglycosides in European data (ECDC data concern all microorganisms isolated from HAI, not only those isolated from BSI); MRSA was 55.6% vs. 34.5% in ECDC data and 50.7% according to NHNS (US data were collected only from central line-associated-BSI infections), MRCNS was 68.9% vs. 85.6% according to ECDC data. Interestingly, in other Polish studies on ICUs, only 13% of MRSA strains were recorded by Wałaszek et al. and 30% by Kołpa et al. [6,9,11,13].

Methicillin resistance in CNS is usually very high; it was detected in 75% of strains in the USA [13] and in Finland [40], in 93.1% in Brazil [41] and in 93.6% in China [42]. However, resistance to macrolides is also often observed, being 63% in Finland [39], 86.2% in Brazil [41], 92.4% in China [42] and 85.3% in Saudi Arabia [43].

Among Gram-negative bacteria from ICUs, *A. baumannii* was the most resistant—its resistance to carbapenems was 70.6% vs. 79.8% in the ECDC data and 46.6% in the NHNS data and 78.8% in the study by Wałaszek et al. Other Polish studies show resistance to carbapenems in *A. baumannii* at 72–87% [7,8,9]. Carbapenem-resistant *P. aeruginosa* was 66.7% vs. 33.4% in the ECDC data and 25.8% in the NHNS data, and *K. pneumoniae* with the ESBL mechanism was 72.4% while ECDC reports 53% of cephalosporin resistance and 89.7% of penicillin resistance. In the study by Wałaszek et al., *K. pneumoniae* with the ESBL mechanism amounted to 76% and there was an equally large number of *E. coli* with ESBL, while in our study it was only 32% [6,11,13].

Resistance of BSI strains from surgical departments was quite similar, compared to ICU: there were more *S. aureus* with the MLS_B_ mechanism (66.7%) than MRSA (33.3%), and more carbapenem-resistant *A. baumannii* (100%) and *E. cloacae* (100%).

There are some limitations associated with this laboratory-based study. Firstly, the demographic information on the study population is limited, and many important clinical data are missing; invasive procedures, antimicrobial usage, co-morbidity, disability, and patient outcome data were not available. Some of the cultured bacteria belonging to the CNS genus *Corynebacterium* and *Clostridium*, could be contaminated, unfortunately we were not able to verify this with clinical data. Resistance to individual antibiotics was not analyzed but only the resistance mechanisms.

In conclusion, the overall microbiology of healthcare-associated BSIs in Polish hospitals was similar to the rates reported for other European countries and for the United States. However, the prevalence of *S. aureus* and yeasts was smaller, and in contrast, CNS was higher. The prevalence of antibiotic susceptibility among Gram-positive microorganisms from ICUs, especially MRSA and HLAR enterococci, was significantly higher. The prevalence of ESBL-producing *K. pneumoniae* was alarmingly high. The use of modern diagnostic techniques, such as MALDI-TOF, guarantees reliable diagnosis, improves patient safety and contributes to the improvement of the effectiveness of therapy.

## Figures and Tables

**Table 1 ijerph-18-02785-t001:** Number of patients, bottle blood culture negativity and positivity rates, by wards, 2016–2019.

Hospital Unit	Patients N (%)	Negative Results of Blood Cultures N (%)	LC-BSI N (%)
ICUs	363 (4.1)	84 (23.1)	279 (76.9)
Surgical units	557 (6.3)	175 (31.4)	382 (68.6)
Other units	7979 (89.7)	4422 (55.4)	3557 (44.6)
Total	8899 (100.0)	4681 (52.6)	4218 (47.4)

Legend: ICU—intensive care unit, LC-BSI—laboratory confirmed bloodstream infection.

**Table 2 ijerph-18-02785-t002:** Share of Gram-positive and Gram-negative bacteria as well as fungi in LC-BSIs by wards, 2016–2019.

	ICUs		Surgical Units		Other Units		Total Units	
	No	Prevalence %	No	Prevalence %	No	Prevalence %	No	Prevalence %
Gram-positive	176	63.1	265	69.4	2548	71.6	2989	70.9
Gram-negative	88	31.5	102	26.7	982	27.6	1 172	27.8
Yeast/Fungi	15	5.4	15	3.9	27	0.8	57	1.4
Total	279	100.0	382	100.0	3557	100.0	4218	100.0

Legend: ICU—intensive care unit, LC-BSI—laboratory confirmed bloodstream infection.

**Table 3 ijerph-18-02785-t003:** Number of different genera and species isolated from LC-BSIs from individual units, 2016–2019.

	Type of Studied Populations			
	ICUs	Surgical Units	Other Units	Total
Gram-positive, number of				
genera	14	20	41	44
species	41	48	138	151
Gram-negative, number of				
genera	11	16	35	38
species	12	23	63	67
Yeast/Fungi, number of				
genera	3	2	1	4
species	5	5	6	9

Legend: ICU—intensive care unit, LC-BSI—laboratory confirmed bloodstream infection.

**Table 4 ijerph-18-02785-t004:** The share of individual species in the total number of LC-BSIs—the table includes only those genera and species that were isolated more than 10 times over the study period: 2016–2019.

	ICUs	Surgical Units	Other Units	Total
	Prevalence %	Prevalence %	Prevalence %	Prevalence %
Gram-positive				
*Clostridium*	1.1	0.0	0.4	0.4
*C. perfringens*	0.4	0.0	0.3	0.3
*Corynebacterium*	3.6	2.9	2.2	2.3
*C. afermentans*	0.4	1.6	0.7	0.8
*C. striatum*	1.4	0.8	0.1	0.3
*Enterococcus*	7.5	5.5	4.4	4.7
*Enterococcus faecalis*	5.0	3.9	3.5	3.7
*Enterococcus faecium*	2.5	1.6	0.8	1.0
*Micrococcus*	1.4	2.1	2.4	2.3
*Micrococcus luteus*	1.4	1.8	1.7	1.7
*Micrococcus* spp.	0.0	0.3	0.6	0.5
*Propionibacterium*	3.6	0.8	0.7	0.9
*Propionibacterium acnes*	3.6	0.8	0.7	0.9
*Staphylococcus*	38.0	49.0	51.4	50.3
*S. aureus*	3.2	5.5	6.9	6.5
*S. capitis*	3.2	0.0	1.9	1.8
*S. epidermidis*	18.6	19.6	19.4	19.4
*S. haemolyticus*	6.1	7.9	5.4	5.7
*S. hominis*	5.4	13.6	15.4	14.6
*S. pettenkoferi*	0.36	0.5	0.7	0.7
*S. warneri*	0.0	0.8	0.7	0.7
*Streptococcus*	5.4	3.9	6.5	6.2
*Str. agalactiae*	0.0	0.5	0.6	0.5
*Str. anginosus*	0.4	0.0	0.3	0.3
*Str. gallolyticus*	0.0	0.0	0.4	0.3
*Str. mitis*	1.1	0.8	0.6	0.7
*Str. oralis*	0.4	0.8	1.0	0.9
*Str. parasanguinis*	0.4	0.3	0.7	0.6
*Str. pneumoniae*	0.4	0.0	1.0	0.9
*Str. pyogenes*	0.7	0.3	0.3	0.3
*Str. salivarius*	0.7	0.3	0.5	0.5
Gram-negative				
*Acinetobacter*	6.1	2.1	1.0	1.4
*Acinetobacter baumannii*	6.1	1.6	0.6	1.0
*Bacteroides*	0.4	1.0	0.8	0.8
*Bacteroides fragilis*	0.4	0.5	0.6	0.6
*Enterobacter*	0.0	0.5	0.9	0.8
*Enterobacter cloacae*	0.0	0.5	0.9	0.8
*Escherichia*	9.0	9.7	14.5	13.7
*Escherichia coli*	9.0	9.7	14.5	13.7
*Klebsiella*	10.8	6.5	5.7	6.1
*Klebsiella oxytoca*	0.4	0.0	0.4	0.4
*Klebsiella pneumoniae*	10.4	6.5	5.3	5.7
*Morganella*	0.7	0.3	0.3	0.3
*Morganella morganii*	0.7	0.3	0.3	0.3
*Proteus*	0.4	1.3	1.3	1.3
*Proteus mirabilis*	0.4	1.3	1.3	1.3
*Pseudomonas*	2.2	1.8	0.9	1.1
*Pseudomonas aeruginosa*	2.2	1.6	0.8	1.0
*Salmonella*	0.4	0.5	0.4	0.5
*Salmonella* spp.	0.4	0.3	0.3	0.3
Yeast strains				
Candida	4.7	3.7	0.8	1.3
*Candida albicans*	2.2	1.0	0.4	0.6
*Candida glabrata*	2.2	1.0	0.1	0.4
Total LC-BSI	100.0	100.0	100.0	100.0

Legend: ICU—intensive care unit, LC-BSI—laboratory confirmed bloodstream infection; the table only includes the genera and species that were detected in blood cultures more than 10 times over the study period (2016–2019). All isolated species are found in the Appendix A.

**Table 5 ijerph-18-02785-t005:** Rare pathogens isolated from LC-BSIs during the study period, 2016–2019.

	Human Microflora	Environmental Species	Veterinary Pathogen or Food Pathogen
Gram-positive			
*Aerobic/* *Microaerophilic*	*Actinomyces neuii* (1) *Brevibacterium paucivorans* (1) *Brevibacterium casei* (4) *Brevibacterium ravenspurgenes* (2) *Dermabacter hominis* (6) *Globicatella sanguinis* (1) *Kocuria rhizophila* (3) *Kocuria kristinae* (5) *Nosocomiicoccus massiliensis* (1) *Rothia dentocariosa* (3) *Rothia mucilaginosa* (9)	*Aerococcus urinae* (1) *Aerococcus viridans* (4) *Arthrobacter cumminsii* (4) *Arthrobacter polychromogenes* (1) *Dietzia cinnamea* (1) *Microbacterium aurum* (1) *Microbacterium lacticum* (1) *Paenarthrobacter ilicis* (1) *Pseudoglutamicibacter cumminsii* (1) *Psychrobacillus psychrotolerans* (1) *Rothia endophytica* (1) *Zimmermanella faecalis* (1)	*Erysipelothrix* *rhusiopathiae* (1) *Macrococcus* *caseolyticus* (1)
Strict anaerobic/ Facultative anaerobic	*Actinomyces odontolyticus* (1) *Actinomyces europaeus* (1) *Actinomyces radicidentis* (1) *Anaerococcus hydrogenalis* (1) *Anaerococcus lactolyticus* (1) *Anaerococcus vaginalis* (1) *Atopobium rimae* (1) *Cutibacterium avidum* (1) *Eubacterium tenue* (1) *Facklamia hominis* (1) *Finegoldia magna* (4) *Gemella haemolysans* (4) *Gemella morbillorum* (1) *Granulicatella adiacens* (3) *Peptoniphilus harei* (3) *Ruminococcus gnavus* (1) *Solobacterium moorei* (1) *Trueperella bernardiae* (1)	*Paenibacillus timonensis* (1) *Paenibacillus xylanilyticus* (1)	*Carnobacterium divergens* (1)
Gram-negative			
Aerobic/ Microaerophilic	*Neisseria flavescens* (2) *Neisseria mucosa* (1) *Pantoea septica* (1) *Psychrobacter phenylpyrovicus*(1) *Psychrobacter sanguinis* (1) *Roseomonas mucosa* (1)	*Aureimonas altamirensis* (1) *Chryseobacterium* sp. (1) *Flavobacterium flavescens* (1) *Sphingobium* sp. (1) *Ochrobactrum anthropi* (1)	*Capnocytophaga canimorsus* (1)
Strict anaerobic/ Facultative anaerobic	*Dialister micraerophilus* (1) *Eikenella corrodens* (1) *Leptotrichia hofstadii* (1) *Parabacteroides distasonis* (1)	*Aeromonas hydrophila* (1)	

Legend: LC-BSI—laboratory confirmed bloodstream infection; number in brackets = number of times isolated during the study period. All isolated species are found in the Appendix A.

**Table 6 ijerph-18-02785-t006:** Mechanisms of resistance in bacteria isolated from LC-BSI during the study period 2016–2019.

	ICUs	Surgical Units	Other Units	Total
	Prevalence %	Prevalence %	Prevalence %	Prevalence %
*E. faecalis*	14	15	126	155
HLAR	84.6	60.0	44.1	49.0
*E. faecium*	7	6	28	41
HLAR	57.1	83.3	46.4	53.7
VRE	14.3	0.0	0.0	2.4
HLAR + VRE	0.0	50.0	14.3	17.1
*S. aureus*	9	21	244	274
MRSA	55.6	33.3	9.4	12.8
MLS_B_	55.6	66.7	23.4	27.7
MRSA + MLS_B_	44.4	33.3	9.0	12.0
all CNS	97	165	1583	1845
MRCNS	71.1	86.1	70.1	71.5
MLS_B_	69.1	81.9	70.0	71.0
MRCNS + MLS_B_	62.9	71.5	57.2	58.8
*Streptococcus*	15	15	231	261
MLS_B_	13.3	40.0	18.6	19.5
*A. baumannii*	17	6	20	43
carbapenem resistant *	70.6	100.0	40.0	60.5
MDR **	52.9	33.3	15.0	32.5
*Pseudomonas aeruginosa*	6	6	29	41
carbapenem resistant *	66.7	16.7	17.2	24.4
MDR **	16.7	0.0	0.0	2.4
*E. cloacae*	0	2	32	34
ESBL	0.0	100.0	56.3	58.8
MDR **	0.0	0.0	6.25	5.9
*E.coli*	25	37	516	578
ESBL	32.0	37.8	25.8	26.8
AmpC	0.0	2.7	0.4	0.5
combined ESBL + AmpC	0.0	0.0	0.2	0.2
MDR **	4.0	2.7	1.6	1.7
*Klebsiella pneumoniae*	29	25	187	241
ESBL	72.4	68.0	61.5	63.5
combined ESBL + AmpC	6.9	8.0	2.1	3.3
MDR **	13.8	4.0	8.0	8.3
other Enterobacterales	8	7	91	106
ESBL	37.5	0.0	9.9	11.3
MDR **	12.5	14.3	5.5	6.6

* carbapenem resistance, resistance to three carbapenems (imipenem, meropenem and ertapenem); ** MDR, multidrug resistance, resistance to 2 or more antimicrobial classes (beta lactams, aminoglycosides, fluoroquinolones, etc.) Legend: ICU—intensive care unit, LC-BSI—laboratory confirmed bloodstream infection, VRE—vancomycin resistant enterococci, HLAR—high-level resistance to aminoglycosides, MRSA—methicillin resistant *S. aureus*, MLS_B_—resistance to macrolides, lincosamides and streptogramins B, CNS—coagulase-negative staphylococci, MRCNS—methicillin resistant coagulase-negative staphylococci, ESBL—extended-spectrum β-lactamase production, AmpC—acquired AmpC β-lactamase production.

## Data Availability

The data presented in this study are available on request from the corresponding author.

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
