# Peer review of "Healthcare-Associated Laboratory-Confirmed Bloodstream Infections—Species Diversity and Resistance Mechanisms, a Four-Year Retrospective Laboratory-Based Study in the South of Poland"

_ijerph, 2021, doi:10.3390/ijerph18052785_

Round 1
Reviewer 1 Report
Materials & Methods
Please describe the number of blood culture bottles per test.
This is very important for assessing the results of blood culture tests.
Two or more cultures are required to determine if the detected bacteria are due to contamination.
If contamination bacteria such as CNSs grow in only one bottle, it is considered likely to be contaminated and can be excluded from the tabulation.
Results & Discussion
Clostridium sp., Corynebacterium sp., and CNSs are also well-known contamination bacteria in blood cultures.
Infection by catheter insertion is also possible, but it is very doubtful whether all of these can be regarded as LC-BSI.
Please spell out abbreviations such as HLAR and LC-BSI when they first appear.
Author Response
Author's Reply to the Review Report
Materials & Methods: Please describe the number of blood culture bottles per test. This is very important for assessing the results of blood culture tests. Two or more cultures are required to determine if the detected bacteria are due to contamination. If contamination bacteria such as CNSs grow in only one bottle, it is considered likely to be contaminated and can be excluded from the tabulation.
Results & Discussion: Clostridium sp., Corynebacterium sp., and CNSs are also well-known contamination bacteria in blood cultures. Infection by catheter insertion is also possible, but it is very doubtful whether all of these can be regarded as LC-BSI.
Authors’ reply: Samples were taken in duplicate (right and left hand) from approximately 70% of the patients. The remaining 30% of patients were sampled in one replicate because, for various reasons (including the patient's condition), it was not possible to collect more samples.
Unfortunately, we are currently unable to obtain information about which specific samples were taken in one repetition so that we can consider whether there is a risk of contamination in their case.
All cultured bacteria included in the study were entered as positive and presented in this form to the doctor who ordered the test.
To the section "limitations" indicated at the end of the article, information was added that contamination may also be partially included in the results : „Some of the cultured bacteria belonging to the CNS, genus Corynebacterium and Clostridium, could be contaminated, unfortunately we were not able to verify this with clinical data.”
Please spell out abbreviations such as HLAR and LC-BSI when they first appear.
Authors’ reply: Corrected.
Reviewer 2 Report
The subject of the original article undertaken by the researchers is up-to-date and important from a clinical point of view. The researchers managed to collect very rich material, which amounted to nearly 9,000 samples. An additional advantage of the study was taking into account the cultivation of both aerobic and anaerobic microorganisms, as well as those clinically significant and with a still poorly described characteristics (mainly environmental strains). Literature is also up-to-date and amounts to over 70% for articles published in the last 5 years.
Below I present a few minor adjustments, the inclusion of which will improve the quality of the article:
- “Resistance mechanisms were investigated according to EUCAST recommendations, different methods were used.” -> Resistance mechanisms were investigated according to EUCAST recommendations” (without the last part of this sentence) [lines 17-18]
- “… coagulase-negative staphylococci, Escherichia coli (13.7%) was the most frequent Gram-negative bacterium.” -> … coagulase-negative staphylococci, while Escherichia coli (13.7%) was the most frequent Gram-negative bacterium [lines 21-22]
- “The highest prevalence faecalis with HLAR mechanism was in ICUs, 84.6%, while E.faeciu-HLAR in surgery, 83.3%” -> The highest prevalence E. faecalis with HLAR mechanism was in ICUs (84.6%), while E. faecium-HLAR in surgery (83.3%) (several corrections here) [lines 25-26]
- “Molecular tests based on PCR and FISH can help increase the sensitivity of detecting bacteria in the blood and also speed up this stage, but they are still not routine diagnostics in Poland [4].” -> Please attach the sentence to the previous paragraph [lines 50-52]
- “genus Acinetobacter (from 17.3% to 34.5%)” -> genus Acinetobacter (from 17.3% to 34.5%) (please use italics) [line 70]
- “Therefore, it is essential to study …” -> Thus, it is essential to study … [line 76]
- “PSM-mec is a surface peptide and the proportion of MRSA with PSM-mec is variable, depending on regional epidemiology. The absence of this peak does not mean that the respective strain is not MRSA, but, if detected, it can be reliably considered as MRSA.” -> it is worth to add a reference informing about this phenomenon here [lines 108-110]
- I believe some pictures/ charts are missing in the Results section. The results presented in the tables are very good (because they present all the results in detail), but for the readers who would like to have a quick overview of the situation, this analysis may be difficult. It is worth considering it, e.g., for Tables 4 and 6
- “Ruminoccus gnavus” -> Ruminococcus gnavus [line 276]
- I believe that the limitations should come before the conclusions - please shift this [line 343-347]
Author Response
Author's Reply to the Review Report
The subject of the original article undertaken by the researchers is up-to-date and important from a clinical point of view. The researchers managed to collect very rich material, which amounted to nearly 9,000 samples. An additional advantage of the study was taking into account the cultivation of both aerobic and anaerobic microorganisms, as well as those clinically significant and with a still poorly described characteristics (mainly environmental strains). Literature is also up-to-date and amounts to over 70% for articles published in the last 5 years.
Authors’ reply: Thank you for this comment!
Below I present a few minor adjustments, the inclusion of which will improve the quality of the article:
- “Resistance mechanisms were investigated according to EUCAST recommendations, different methods were used.” -> Resistance mechanisms were investigated according to EUCAST recommendations” (without the last part of this sentence) [lines 17-18]
- “… coagulase-negative staphylococci, Escherichia coli(13.7%) was the most frequent Gram-negative bacterium.” -> … coagulase-negative staphylococci, while Escherichia coli (13.7%) was the most frequent Gram-negative bacterium [lines 21-22]
- “The highest prevalence faecaliswith HLAR mechanism was in ICUs, 84.6%, while faeciu-HLAR in surgery, 83.3%” -> The highest prevalence E. faecalis with HLAR mechanism was in ICUs (84.6%), while E. faecium-HLAR in surgery (83.3%) (several corrections here) [lines 25-26]
- “Molecular tests based on PCR and FISH can help increase the sensitivity of detecting bacteria in the blood and also speed up this stage, but they are still not routine diagnostics in Poland [4].” -> Please attach the sentence to the previous paragraph [lines 50-52]
- “genus Acinetobacter (from 17.3% to 34.5%)” -> genus Acinetobacter(from 17.3% to 34.5%) (please use italics) [line 70]
- “Therefore, it is essential to study …” -> Thus, it is essential to study … [line 76]
Authors’ reply: Corrected all above.
- “PSM-mec is a surface peptide and the proportion of MRSA with PSM-mec is variable, depending on regional epidemiology. The absence of this peak does not mean that the respective strain is not MRSA, but, if detected, it can be reliably considered as MRSA.” -> it is worth to add a reference informing about this phenomenon here [lines 108-110]
Authors’ reply: Reference was added: Hu Y, Huang Y, Lizou Y, Li J, Zhang R. Evaluation of Staphylococcus aureus Subtyping Module for Methicillin-Resistant Staphylococcus aureus Detection Based on Matrix-Assisted Laser Desorption Ionization Time-of-Flight Mass Spectrometry. Front Microbiol. 2019;10:2504.
- I believe some pictures/ charts are missing in the Results section. The results presented in the tables are very good (because they present all the results in detail), but for the readers who would like to have a quick overview of the situation, this analysis may be difficult. It is worth considering it, e.g., for Tables 4 and 6
Authors’ reply: All study details are included for publication as Supplementary Material. To simplify Tables 4 and 6 to make it easier for quick review, the columns with the number of individual bacteria were removed, leaving the columns showing prevalence (%).
- “Ruminoccus gnavus” -> Ruminococcus gnavus[line 276]
- I believe that the limitations should come before the conclusions - please shift this [line 343-347]
Authors’ reply: Corrected.